# Changes in PHQ-9 depression scores in acute stroke patients shortly after returning home

**Brent Strong**[1], **Michele C. Fritz**[1], **Liming Dong**[2], **Lynda D. Lisabeth**[2], **Mathew J. Reeves**[1]*

1 Department of Epidemiology and Biostatistics, College of Human Medicine, Michigan State University, East Lansing, Michigan, United States of America, 2 Department of Epidemiology, School of Public Health, University of Michigan, Ann Arbor, Michigan, United States of America

* reevesm@msu.edu

**Data Availability Statement:** https://osf.io/4hzdf/ DOI 10.17605/OSF.IO/4HZDF.

**Funding:** Grant support for MISTT (Michigan Stroke Transitions Trial) was awarded to corresponding author MJR by the Patient-Centered

## Abstract

### Introduction

Post-stroke depression is a disabling condition that occurs in approximately one-third of stroke survivors. There is limited information on changes in depressive symptoms shortly after stroke survivors return home. To identify factors associated with changes in post-stroke depressive symptoms during the early recovery period, we conducted a secondary analysis of patients enrolled in a clinical trial conducted during the transition period shortly after patients returned home (MISTT).

### Methods

The Michigan Stroke Transitions Trial (MISTT) tested the efficacy of social worker case management and access to online information to improve patient-reported outcomes following an acute stroke. Patient Health Questionnaire-9 (PHQ-9) scores were collected via telephone interviews conducted at 7 and 90 days post-discharge; higher scores indicate more depressive symptoms. Generalized estimating equations were used to identify independent predictors of baseline PHQ-9 score at 7 days and of changes over time to 90 days.

### Results

Of 265 patients, 193 and 185 completed the PHQ-9 survey at 7 and 90 days, respectively. The mean PHQ-9 score was 5.9 at 7 days and 5.1 at 90 days. Older age, being unmarried, and having moderate stroke severity (versus mild) were significantly associated with lower 7-day PHQ-9 scores (indicating fewer depressive symptoms). However, at 90 days, both unmarried patients and those with moderate or high stroke severity had significant increases in depressive symptoms over time.

### Conclusions

In stroke patients who recently returned home, both marital status and stroke severity were associated with depressive symptom scores; however, the relationships were complex. Being unmarried and having higher stroke severity was associated with fewer depressive

Outcomes Research Institute ([PCORI.org](PCORI.org)) Award No. IHS-1310-07420-01. PCORI had no direct role in the design and conduct of the study; collection, management, analysis, and interpretation of the data; preparation, review, or approval of this article; or in the decision to submit this article for publication.

**Competing interests:** The authors have declared that no competing interests exist.

symptoms at baseline, but both factors were associated with worsening depressive symptoms over time. Identifying risk factors for changes in depressive symptoms may help guide effective management strategies during the early recovery period.

## Introduction

Despite the long term decreases in stroke mortality, the burden of stroke-related disability remains high [1]. Depression is a disabling neuropsychiatric sequela that occurs in one-third of stroke survivors. and has been shown to interfere with recovery [2], worsen caregiver burden [3], and increase the risk of recurrent stroke, rehospitalization, and institutionalization [4, 5]. Clinical stroke guidelines recommend regular screening of stroke patients for depression and the administration of pharmacotherapy when indicated, although the optimal timing for such screening remains unclear [6]. The prevalence of post-stroke depression has been well studied in various settings; a 2014 meta-analysis found that the pooled prevalence of depression was 28% at 0 to 1 month after stroke onset and 36% at 2 to 5 months after stroke onset [7]. However, data identifying factors associated with changes in depressive symptoms are limited, especially during the early recovery period shortly after stroke survivors have returned home following hospitalization [3, 8, 9]. Identifying factors associated with early changes in depressive symptoms may improve screening for post-stroke depression and help identify suitable candidates for prophylactic treatment with antidepressants [10].

The Michigan Stroke Transitions Trial (MISTT) was designed to investigate the effects of two transitional care interventions—a home-based social worker-led case management program alone or combined with a website providing stroke-related information—on patient-reported outcomes 90 days post discharge [11, 12]. The objectives of this secondary analysis of the MISTT study data are to: (1) describe depressive symptom severity in acute stroke patients 7 and 90 days after returning home; and (2) identify factors associated with depressive symptom severity at 7 days and with changes in symptom severity between 7 and 90 days.

## Methods

### Design

Details regarding the MISTT study design and protocol [12] and primary patient results [11] have been previously published. MISTT was an open (non-blinded), randomized, parallel, 3-group clinical trial registered at clinicaltrials.gov (NCT02653170). Two hundred sixty-five participants were recruited from 3 Michigan hospitals between January 2016 and July 2017. Patients were randomized to one of 3 treatment groups on the day they were discharged home (either directly from the hospital or following a 4-week or shorter stay in a rehabilitation facility). The treatment groups were usual care (UC), home-based social work case management (SWCM) program [13], or SWCM program plus access to the MISTT website, a curated patient-orientated information resource. The average duration of case management activities during the trial was 63 days [11].

### Participants

Eligible participants had a confirmed diagnosis of acute ischemic or hemorrhagic stroke, were living at home prior to admission, and were discharged either directly home or to an acute or sub-acute rehabilitation facility but then returned home within one month. Patients had to

exhibit functional deficits (modified Rankin Scale [mRS] score ≥1) or have rehabilitative therapy recommended at discharge. Proxy consent was obtained for patients who failed cognitive screening or had significant stroke-related communication deficits [12]. Patients also had the option of identifying a caregiver (defined as the person most likely to be helping them at home) who could also enroll in the trial [12]. The MISTT study was approved by the Biomedical & Health Institutional Review Board at Michigan State University and by IRBs at each study site [12]. Written informed consent was obtained from all enrolled participants.

## Outcomes and data collection

Trained study coordinators at each hospital abstracted or collected information on patient demographics, living arrangements, pre-stroke function, past medical history (including history of depression), clinical features, treatments, and discharge instructions [11, 12]. Stroke severity was classified using the National Institutes of Health Stroke Scale [NIHSS] or the Glasgow Coma Scale (GCS) if NIHSS was not available for cases of hemorrhagic stroke. Stroke severity was categorized into 3 groups: mild (NIHSS 1–5 or GCS 13–15), moderate (NIHSS 6–13 or GCS 5–12), or severe (NIHSS 14–42 or GCS 3–4). Information on marital status was obtained from the 7-day telephone interview (described below). For 34 subjects who did not complete a 7-day interview we imputed their marital status to 'married' if a caregiver was interviewed who was a spouse (n = 16), 'single' if the caregiver was not a spouse (n = 11), and 'married' (the most common category) for the remainder (n = 7) who lacked a caregiver.

After discharge, trained telephone interviewers collected patient-reported outcomes data including PHQ-9 scores at 7 days (range 5–21 days) and 90 days (range 83–111 days) after the patients returned home. The PHQ-9 questionnaire collects information on the extent to which the participant has been bothered by different depressive symptoms including the absence of interest/pleasure, feeling down, disturbances in sleep, feeling tired, changes in appetite, feeling bad about oneself, trouble concentrating, changes in energy level and suicidal ideation [14]. Each item was scored between 0 (not at all) and a maximum of 3 (nearly every day) with total scores summed to a range of 0–27 and higher scores indicating greater symptom severity. The 7-day interview referenced the frequency of the symptoms in the last week (since returning home), while the 90-day questionnaire referred to the last two weeks. A PHQ-9 score of 10 or greater has been shown to be a sensitive and specific cut-off for the presence of clinical depression among stroke patients [15]. The primary analysis of the MISTT trial found no statistically significant intervention effects on PHQ-9, which was defined as a secondary outcome [11]; thus, for this analysis we examined predictors of baseline scores and change over time using a cohort design framework. The intervention variable (randomization group) was retained in all models. We used the STROBE guidelines to report our findings [16]; a completed checklist is included in the Table A in S1 File.

## Statistical analysis

PHQ-9 score data (symptom severity) collected at 7 and 90 days was analyzed as a continuous outcome using generalized estimating equations (GEE) (PROC GENMOD, SAS Version 9.4) with an identity link function and robust standard errors to account for within-subject correlation [17]. The following variables were pre-specified as potential cofounders: age, race, sex, hospital discharge destination (i.e., home, acute rehabilitation, or sub-acute rehabilitation), and randomization arm (see S1 File). These five variables were included in all multivariable models regardless of statistical significance. Other predictor variables were selected for inclusion in the multivariable model building phase if they had an unadjusted statistical association with PHQ-9 score of P<0.20. To generate parameter estimates for both the baseline (7-day)

score and the change in score over time (i.e., 90-day minus 7-day) we included 2-way interaction terms between time (7 or 90 days) and each predictor. To ensure the parsimony of the final multivariable model, backwards variable selection methods were used to drop predictors that were not statistically significant (P<0.05) after adjustment. All 2-way interactions involving time were retained in the final model regardless of statistical significance, thereby allowing an assessment of the effect of each predictor variable on baseline score and change over time.

We conducted several sensitivity analyses. In our primary analysis we chose not to adjust for history of ever having been diagnosed with depression (abstracted from patient medical records), as this variable could mask other important predictors of depressive symptoms and is likely an underestimate of the true pre-stroke prevalence of depression in our study population. However, we added history of depression and its interaction with time to the final model to determine if it changed the significance of the identified predictors. We also tested interactions between important predictor variables (namely, stroke severity and marital status) and age, sex, and race. Although we describe the general conclusions of any significant interactions, we did not incorporate these interactions into the final model because they were exploratory, and the sample size was too small to make reliable inferences about their direction and magnitude. Finally, we explored the effect of dropping subjects whose PHQ-9 scores were collected through proxy and accounted for missing PHQ-9 data in the multivariable model using inverse probability weighting (IPW) [18–20]. Additional details regarding the IPW analyses are provided in the S1 File.

## Results

Of 265 patients enrolled in MISTT, PHQ-9 data was available for 193 (73%) at 7 days, and 185 (70%) at 90 days. A total of 214 patients (81%) completed at least one PHQ-9 survey, and 164 (62%) completed both for a total of 378 observations. We examined differences between patients with and without complete PHQ-9 data; the only significant differences (P<0.05) were a higher prevalence of non-white race and longer acute hospital stays among patients missing data at 7 days (Table B in S1 File), and a higher prevalence of non-white race, longer acute hospital stays, lower prevalence of prior stroke, and lower cognitive ability among patients missing data at 90 days (Table C in S1 File). Unadjusted mean PHQ-9 scores at 7 and 90 days by patient characteristics are presented in Table 1. The overall mean PHQ-9 score was 5.9 at 7 days and 5.1 at 90 days, while the proportion of subjects with clinical depression (PHQ-9 score ≥ 10) was 21% at 7 days, and 16% at 90 days. Statistically significant differences in 7-day scores were observed for sex, race, and history of depression, while significant differences in 90-day scores were observed for age, marital status, and history of depression.

Results of the final GEE multivariable linear model for baseline (7-day) PHQ-9 scores are shown in Table 2. In addition to the pre-specified confounders (i.e., age, race, sex, discharge destination, and randomization arm), only the main effects of stroke severity and marital status were retained (based on P<0.05) in the final multivariable model. After adjustment, older age, being unmarried, and having moderate stroke severity (versus mild) were significantly associated with lower baseline PHQ-9 scores, while female sex and discharge to a sub-acute rehabilitation facility were both associated with significantly higher baseline PHQ-9 scores (indicating greater depressive symptoms). The largest decrements were observed in patients older than 80 (mean score 2.4 units lower compared to patients younger than 60), patients with moderate stroke severity (mean score 2.4 units lower compared to mild stroke), and those unmarried (mean score 1.7 units lower compared to married patients). The largest increases were observed for patients discharged to subacute rehabilitation who had a mean score 2.7 units higher that patients discharged home.

**Table 1. Unadjusted mean PHQ-9 score at 7 and 90 days by patient characteristics (n = 214 patients\*).**

| Patient Characteristic | Number of Observations at 7 Days (%) | Mean PHQ-9 Score at 7 Days (Standard Deviation) | Number of Observations at 90 Days (%) | Mean PHQ-9 Score at 90 Days (Standard Deviation) |
|---|---|---|---|---|
| *Overall* | 193 | 5.9 (5.1) | 185 | 5.1 (5.3) |
| *Age¶* | | | | |
| 18–59 | 50 (26) | 6.9 (5.5) | 53 (29) | 6.7 (6.9) |
| 60–69 | 62 (32) | 6.5 (5.6) | 58 (31) | 5.6 (4.6) |
| 70–79 | 52 (27) | 5.0 (3.9) | 48 (26) | 3.5 (3.8) |
| ≥80 | 29 (15) | 4.7 (5.2) | 26 (14) | 3.8 (4.8) |
| *Sex\|\|* | | | | |
| Male | 98 (51) | 5.4 (5.0) | 93 (50) | 4.7 (5.6) |
| Female | 95 (49) | 6.5 (5.2) | 92 (50) | 5.6 (5.1) |
| *Race\|\|* | | | | |
| White | 161 (83) | 5.6 (5.1) | 153 (83) | 4.9 (5.2) |
| Non-white | 32 (17) | 7.7 (5.2) | 32 (17) | 5.9 (5.8) |
| *Marital Status¶* | | | | |
| Married | 113 (59) | 6.4 (5.5) | 111 (60) | 4.4 (5.0) |
| Unmarried | 80 (41) | 5.2 (4.6) | 74 (40) | 6.2 (5.7) |
| *History of Depression \|\|¶* | | | | |
| Yes | 24 (12) | 9.1 (5.7) | 22 (12) | 8.1 (6.3) |
| No | 169 (88) | 5.5 (4.9) | 163 (88) | 4.7 (5.1) |
| *Stroke Type* | | | | |
| Ischemic | 169 (88) | 6.0 (5.2) | 160 (86) | 5.2 (5.3) |
| Hemorrhagic† | 24 (12) | 5.6 (4.7) | 25 (14) | 4.6 (5.7) |
| *Stroke Severity§* | | | | |
| Mild | 141 (73) | 6.5 (5.5) | 136 (74) | 4.6 (4.9) |
| Moderate | 41 (21) | 4.0 (3.5) | 36 (19) | 6.1 (6.2) |
| Severe | 11 (6) | 5.7 (4.1) | 13 (7) | 7.8 (7.0) |
| *Randomization Arm‡* | | | | |
| Usual Care | 60 (31) | 5.9 (5.5) | 58 (31) | 5.0 (4.7) |
| SWCM | 66 (34) | 5.7 (5.0) | 64 (35) | 4.9 (4.8) |
| SWCM + | 67 (35) | 6.1 (5.0) | 63 (34) | 5.4 (6.4) |
| *Discharge Destination* | | | | |
| Home | 88 (46) | 5.8 (4.8) | 87 (47) | 4.1 (4.4) |
| Acute Rehab | 87 (45) | 5.7 (5.2) | 83 (45) | 6.0 (6.3) |
| Subacute Rehab | 18 (9) | 7.4 (6.4) | 15 (8) | 5.9 (4.1) |
| *Modified Rankin Scale Score at Discharge* | | | | |
| 0–2 | 131 (68) | 5.9 (5.0) | 127 (69) | 5.1 (5.4) |
| 3–5 | 62 (32) | 5.9 (5.4) | 58 (31) | 5.1 (5.2) |
| *Length of Stay in the Acute Hospital Setting* | | | | |
| 1–2 days | 55 (28) | 6.0 (5.1) | 53 (29) | 4.8 (5.4) |
| 3–5 days | 91 (47) | 5.3 (5.0) | 85 (46) | 4.5 (4.4) |
| >5 days | 47 (24) | 7.0 (5.5) | 47 (25) | 6.7 (6.6) |
| *Length of Stay in Rehabilitation* | | | | |
| 0 days | 88 (46) | 5.8 (4.8) | 87 (47) | 4.1 (4.4) |
| 1–10 days | 41 (21) | 5.7 (5.6) | 35 (19) | 5.3 (6.0) |
| >10 days | 64 (33) | 6.2 (5.4) | 63 (34) | 6.3 (6.0) |
| *Consented Caregiver* | | | | |
| Yes | 128 (66) | 6.2 (5.3) | 120 (65) | 5.2 (5.6) |

*(Continued)*

**Table 1.** (Continued)

| Patient Characteristic | Number of Observations at 7 Days (%) | Mean PHQ-9 Score at 7 Days (Standard Deviation) | Number of Observations at 90 Days (%) | Mean PHQ-9 Score at 90 Days (Standard Deviation) |
|---|---|---|---|---|
| No | 65 (34) | 5.4 (4.7) | 65 (35) | 4.9 (5.0) |
| *Collection of PHQ-9 Score through Proxy* | | | | |
| Yes | 14 (7) | 8.4 (6.2) | 14 (8) | 6.7 (6.4) |
| No | 179 (93) | 5.7 (5.0) | 171 (92) | 5.0 (5.2) |

* A total of 214 patients had either a 7-day or 90-day PHQ-9 value recorded.

†Hemorrhagic strokes include both intracerebral hemorrhage and subarachnoid hemorrhage.

‡ SWCM indicates the social worker case management arm, and SWCM+ indicates the social worker case management plus website arm.

§ Stroke severity was categorized as mild (National Institutes of Health Stroke Scale [NIHSS] 1–5, or Glasgow Coma Scale [GCS] 13–15), moderate (NIHSS 6–13, or GCS 5–12), and severe (NIHSS 14–42, or GCS 3–4).

|| Variable is significantly associated with 7-day PHQ-9 scores (P<0.05). Statistical significance was determined with a Wilcoxon two-sample test for variables with two levels and a Kruskal-Wallis test for variables with 3 or more levels.

¶ Variable is significantly associated with 90-day PHQ-9 scores (P<0.05). Statistical significance was determined with a Wilcoxon two-sample test for variables with two levels and a Kruskal-Wallis test for variables with 3 or more levels.

In the sensitivity analyses, history of depression was associated with higher PHQ-9 scores at 7 days (β = 3.6 [95% CI = 1.3, 5.8]); when this variable was included in the final multivariable model, it attenuated the effect estimate for female sex which became non-significant (β = 1.1 [95% CI = -0.3, 2.5]), but increased the estimate for non-white race which became statistically significant (β = 2.2 [95% CI = 0.4, 3.9]). The only significant interaction at baseline was between age and marital status (P = 0.045). The observation that unmarried patients had lower 7-day PHQ-9 scores than married patients was limited to the 60 to 69 and 70 to 79 age groups. Conversely, unmarried patients had higher PHQ-9 scores than married patients in the age group over 80. When patients for whom data was collected by proxy were excluded, there were small changes to two variables; non-white race became a significant predictor of higher baseline PHQ-9 scores, while the effect of discharge to subacute rehabilitation became non-significant (Table D in S1 File). After accounting for missing data with IPW, the results of the final model were similar although there were very modest changes to the statistical significance of female sex, discharge destination (sub-acute rehabilitation), and non-white race (Table E in S1 File).

Results from the final multivariable GEE linear model relevant to changes in PHQ-9 score over time are shown in Table 3. The only predictors that had significant interactions with time were marital status (P<0.001) and severity (P<0.001). Being unmarried was associated with an increase in PHQ-9 scores over time (3.2 unit increase compared to married), which was the opposite effect observed at 7 days where being unmarried was associated with lower scores. Also, in contrast to the direction of the relationship observed between stroke severity and 7-day scores, now both moderate stroke severity (3.3 unit increase) and high stroke severity (3.2 unit increase) were associated with increases in PHQ-9 score over time, when compared to patients with mild stroke severity.

In the sensitivity analyses, history of depression was not associated with a change in PHQ-9 score over time, and its inclusion in the final multivariable model did not change the results. When patients whose scores were collected by proxy were excluded, the effect of being unmarried increased (β = 3.9 [95% CI = 2.5, 5.2]) as did the effect of having moderate stroke severity (β = 3.6 [95% CI = 2.0, 5.3]); however, the effect of high stroke severity decreased (β = 1.9 [95% CI = -1.1, 4.8]) (Table F in S1 File). Accounting for missing data using IPW did not meaningfully alter the results of the change over time analysis (Table G in S1 File).

**Table 2. Final multivariable linear regression analysis of baseline PHQ-9 score at 7 days (n = 193 patients).**

| Predictor | Unadjusted β (95% CI)* | Adjusted β (95% CI)* |
|---|---|---|
| *Age* | | |
| 18–59 | 0 (ref) | 0 (ref) |
| 60–69 | -0.2 (-2.2, 1.8) | -0.7 (-2.6, 1.2) |
| 70–79†‡ | -1.8 (-3.6, 0.0) | -2.4 (-4.2, -0.6) |
| ≥80‡ | -2.1 (-4.4, 0.3) | -2.4 (-4.7, -0.1) |
| *Sex* | | |
| Male | 0 (ref) | 0 (ref) |
| Female‡ | 1.3 (-0.2, 2.7) | 1.5 (0.1, 3.0) |
| *Race* | | |
| White | 0 (ref) | 0 (ref) |
| Non-white† | 2.1 (0.2, 3.9) | 1.7 (0.0, 3.5) |
| *Marital Status* | | |
| Married | 0 (ref) | 0 (ref) |
| Unmarried‡ | -1.3 (-2.7, 0.1) | -1.7 (-3.1, -0.3) |
| *Discharge Destination* | | |
| Home | 0 (ref) | 0 (ref) |
| Acute Rehab | 0.2 (-1.3, 1.6) | 0.7 (-0.7, 2.2) |
| Sub-acute Rehab‡ | 1.7 (-1.3, 4.7) | 2.7 (0.1, 5.4) |
| *Randomization Arm*§ | | |
| Usual Care | 0 (ref) | 0 (ref) |
| SWCM | -0.2 (-2.0, 1.6) | -0.1 (-1.8, 1.6) |
| SWCM+ | 0.1 (-1.7, 1.9) | 0.4 (-1.4, 2.1) |
| *Stroke Severity* | | |
| Mild | 0 (ref) | 0 (ref) |
| Moderate†‡ | -2.4 (-3.8, -1.0) | -2.4 (-3.8, -1.0) |
| High | -0.7 (-3.1, 1.7) | -0.7 (-2.9, 1.4) |

*Unadjusted effect estimates were generated by a model including the predictor, time, and their interaction. Adjusted estimates are adjusted for all other predictors included in the table. A positive value indicates an association with more depressive symptoms.

† P-value < 0.05 in unadjusted analyses.

‡ P-value < 0.05 in adjusted analyses.

§ SWCM indicates the social worker case management arm, and SWCM+ indicates the social worker case management plus website arm.

## Discussion

In this secondary analysis of a cohort of stroke patients enrolled in a community-based randomized clinical trial, we found that marital status and stroke severity were consistent predictors of depressive symptoms; however, the observed relationships were complex. While being unmarried and having higher stroke severity were associated with lower depressive symptom scores at baseline, they were both associated with increases in symptom scores between 7 and 90 days, even though they started off with lower values. Specifically, being unmarried was associated with a 3.2 unit increase in PHQ-9 scores over time, compared to married participants. Being unmarried is a well-documented risk factor for depression in the general elderly population [21], although evidence for this association among stroke patients is more limited [22]. Other related social factors and living circumstances including social isolation and living alone have also been shown to be risk factors for post-stroke depression [23]. Although our finding

**Table 3. Final multivariable linear regression analysis of change in PHQ-9 score from 7 days to 90 days (n = 378 observations from 214 unique patients).**

| Predictor | Unadjusted β (95% CI)* | Adjusted β (95% CI)* |
|---|---|---|
| *Age* | | |
| 18–59 | 0 (ref) | 0 (ref) |
| 60–69 | -0.6 (-2.8, 1.6) | 0.0 (2.0, 2.0) |
| 70–79 | -1.2 (-3.3, 0.8) | -0.9 (-2.7, 1.0) |
| ≥80 | -0.4 (-2.7, 1.8) | 0.0 (-2.1, 2.2) |
| *Sex* | | |
| Male | 0 (ref) | 0 (ref) |
| Female | -0.3 (-1.8, 1.1) | -0.5 (-1.9, 0.9) |
| *Race* | | |
| White | 0 (ref) | 0 (ref) |
| Non-white | -1.0 (-3.0, 0.9) | -1.3 (-3.1, 0.5) |
| *Marital Status* | | |
| Married | 0 (ref) | 0 (ref) |
| Unmarried†‡ | 3.1 (1.7, 4.5) | 3.2 (1.9, 4.6) |
| *Discharge Destination* | | |
| Home | 0 (ref) | 0 (ref) |
| Acute Rehab | 1.5 (0.0, 3.0) | 0.7 (-0.7, 2.1) |
| Sub-acute Rehab | 0.2 (-2.6, 2.9) | -1.0 (-3.7, 1.8) |
| *Randomization Arm*§ | | |
| Usual Care | 0 (ref) | 0 (ref) |
| SWCM | -0.1 (-1.8, 1.5) | -0.5 (-2.1, 1.0) |
| SWCM+ | 0.1 (-1.9, 2.0) | -0.2 (-1.9, 1.5) |
| *Stroke Severity* | | |
| Mild | 0 (ref) | 0 (ref) |
| Moderate†‡ | 3.6 (1.8, 5.4) | 3.3 (1.7, 5.0) |
| High†‡ | 3.8 (1.5, 6.0) | 3.2 (0.5, 5.9) |

* Model estimates are based on the inclusion of two-way interactions between time and each predictor. Adjusted estimates are adjusted for all other predictors included in the table. Positive values are indicative of an increase in PHQ-9 scores over time compared to the reference.

† P-value < 0.05 in unadjusted analyses.

‡ P-value < 0.05 in adjusted analyses.

§ SWCM indicates the social worker case management arm, and SWCM+ indicates the social worker case management plus website arm.

that unmarried patients had worsening symptom scores over time is consistent with the prior literature, the finding that unmarried patients had lower depressive symptoms at baseline is not. Both moderate stroke severity and high stroke severity were associated with increases (around 3.2 units) in PHQ-9 score over time relative to mild severity, findings that are also consistent with previous research. It has been demonstrated consistently that higher stroke severity is associated with post-stroke depression [2, 22, 23]. Thus, the finding that patients with greater stroke severity had lower symptom scores at baseline appears to be at odds with the prior research and as such remains an unexpected finding.

We are unsure of the exact reasons for these unexpected findings, although several possibilities exist. The first is selection bias, which can occur when an analysis is conditioned on a variable affected by both the exposure (or cause of the exposure) and outcome (or cause of the outcome) [24]. Participation in the MISTT study required informed consent, which is

plausibly affected by a patient's stroke severity, marital status, and depressive symptoms in the hospital. Thus, selection bias could have been introduced at the point of initial enrollment. Second, these findings may reflect the complex interplay of cognitive and framing effects [25], which can introduce bias in the self-assessment and reporting of complex traits such as depressive symptoms. Finally, it is possible that these findings could be influenced by the phenomenon of response shift where a patient's response is influenced by changes in their viewpoint of a particular construct due to shifts in their internal standards or values [26, 27].

Both the prevalence of post-stroke depression and the average severity of depressive symptoms (i.e., the absolute scores) in our population were relatively modest. We found that the prevalence of clinical depression (based on a PHQ-9 score ≥10) [15], declined from 21% of patients at 7 days to 16% at 90 days. These figures are lower than those reported in a 2014 meta-analysis which found, among 61 observational studies, a pooled prevalence of 28% in the 0–1 month period following stroke onset, and 36% in the 2–5 month period [7]. However, interpretations of these pooled estimates are complicated by wide heterogeneity between studies where estimates of the frequency of post-stroke depression at 3 months ranged from 14% to 63% [28, 29]. The lower prevalence of depression in our study may be reflective of the fact that the data come from a patient population that agreed to participate in a clinical trial. Also, MISTT did not enroll patients who were discharged directly to long term care or were expected to be in rehabilitation for longer than 4 weeks. The net effect of this was that there were fewer patients with severe stroke enrolled in the study [11].

In addition to stroke severity and marital status, our multivariable analysis identified several other predictors of post-stroke depressive symptoms 7 days after discharge home, including older age, female sex, and discharge to subacute rehabilitation. However, only age remained a significant predictor in the sensitivity analyses. Although older age was identified as a risk factor of post-stroke depression in only 3 of 16 studies in a previous systematic review [23], several other studies have documented a lower rate of depression among older patients [28, 30, 31], which was the case in our study.

The strengths of this study include its prospective design, the availability of detailed demographic and clinical data for each patient, the inclusion of all stroke subtypes, and the assessment of changes in PHQ-9 scores early after return home, which is a critical time period that has been under-studied in stroke populations [3, 8, 9]. Our findings that both unmarried patients and patients with a severe stroke had worsening PHQ-9 scores in the early transition period are important because it provides an indication of which stroke patients could be targeted for enhanced screening and treatment including pharmacotherapy or psychosocial intervention. Despite data showing that prophylactic administration of anti-depressants may be effective [10], this treatment is not recommended by current stroke clinical guidelines [6]. Finally, our results were largely robust when we conducted multiple sensitivity analyses. Although the rate of missing data was relatively high, when missing participants were accounted for using IPW, changes in model results were minimal.

There are some limitations to consider including the fact that the data were obtained from a sample of consented participants enrolled in a clinical trial; as discussed previously, selection bias may be at play. In addition, although the MISTT trial was pragmatic and inclusion criteria were not overly restrictive [11], it is likely that the study sample does not fully generalize to the wider stroke population. We also note that the magnitude of the associations for the significant predictors were generally modest and were less than the minimal clinically important difference of 5 points for the PHQ-9 [32]. Finally, comparisons of this study with previous analyses are complicated by the infrequent use of PHQ-9 to investigate factors associated with post-stroke depression. While more recent studies have used the PHQ-8 [33, 34] and a meta-analysis of diagnostic validity studies found that the PHQ-9 performed better than most scales for

post-stroke depression screening [35], two systematic reviews of observational studies both published in 2014 did not include any studies that used the PHQ-9 scale [22, 23].

In summary, we found several predictors of post-stroke depressive symptoms in stroke patients who recently returned home. Being unmarried and having higher stroke severity were associated with fewer depressive symptoms at baseline but worsening depressive symptoms over time. Given the paucity of studies examining factors affecting changes in post-stroke depression early in the recovery period following the return to home, further research is needed to confirm these findings and those of other studies. Identifying risk factors for changes in depressive symptoms may help guide effective management strategies during the early recovery period.

## Supporting information

**S1 File. Supplemental methods, results, and references.**
(DOCX)

## Acknowledgments

We thank the entire MISTT team for their contributions as well as the patients who participated and made this study possible.

## Author Contributions

**Conceptualization:** Mathew J. Reeves.

**Data curation:** Michele C. Fritz.

**Formal analysis:** Brent Strong.

**Funding acquisition:** Mathew J. Reeves.

**Methodology:** Michele C. Fritz, Liming Dong, Mathew J. Reeves.

**Project administration:** Michele C. Fritz, Mathew J. Reeves.

**Supervision:** Michele C. Fritz, Lynda D. Lisabeth, Mathew J. Reeves.

**Writing – original draft:** Mathew J. Reeves.

**Writing – review & editing:** Brent Strong, Lynda D. Lisabeth.

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
