## [Decision Letter · Decision Letter 0]

9 Aug 2021

PONE-D-21-15011

Changes in PHQ-9 Depression Scores in Acute Stroke Patients Shortly After Returning Home

PLOS ONE

Dear Dr. Reeves,

Thank you for submitting your manuscript to PLOS ONE. I apologise for the length of time between submission and reviews. After careful consideration, we feel that it has merit but does not fully meet PLOS ONE’s publication criteria as it currently stands. Therefore, we invite you to submit a revised version of the manuscript that addresses the points raised during the review process.

We look forward to receiving your revised manuscript.

Kind regards,

Seana Gall

Academic Editor

PLOS ONE

1. Please ensure that your manuscript meets PLOS ONE's style requirements, including those for file naming. The PLOS ONE style templates can be found at https://journals.plos.org/plosone/s/file?id=wjVg/PLOSOne_formatting_sample_main_body.pdf and https://journals.plos.org/plosone/s/file?id=ba62/PLOSOne_formatting_sample_title_authors_affiliations.pdf.

Additional Editor Comments (if provided):

Reviewers' comments:

Reviewer's Responses to Questions

**Comments to the Author**

1. Is the manuscript technically sound, and do the data support the conclusions?

Reviewer #1: Yes

Reviewer #2: Yes

2. Has the statistical analysis been performed appropriately and rigorously? 

Reviewer #1: Yes

Reviewer #2: Yes

3. Have the authors made all data underlying the findings in their manuscript fully available?

Reviewer #1: Yes

Reviewer #2: Yes

4. Is the manuscript presented in an intelligible fashion and written in standard English?

Reviewer #1: Yes

Reviewer #2: Yes

5. Review Comments to the Author

Reviewer #1: Thank you for the opportunity to review this manuscript. I found the manuscript interesting, clearly presented and well explained. The objective of the study is to identify the factors associated with the severity of depressive symptoms at 7 days and 90 days post-discharge. The authors performed a secondary analysis of patients enrolled in the MISTT trial.

Question for the authors:

• What is your reason for choosing to measure depressive symptoms at 7 days and 90 days post-discharge? Is there any evidence in the literature that shows stroke survivors develop depressive symptoms as early as 7 days? or within 7-90 days post-discharge? I think it will be good to have this justification.

Reviewer #2: The authors have presented an interesting study regarding the changes in PHQ-9 in acute stroke patients shortly after returning home and reported that being unmarried and having higher stroke severity was associated with fewer depressive symptoms at baseline, but both factors were associated with worsening depressive symptoms over time. I have just a few queries:

1. Could the authors please explain the reason for line in Results section (quoted below) where moderate and mild stroke are compared but not severe stroke?

“After adjustment, older age, being unmarried, and having moderate stroke severity (versus mild) were significantly associated with lower baseline PHQ-9 scores,…”

2. It is mentioned that female sex is an independent predictor of the outcomes. Did authors examine any interaction of sex with severity of stroke that could add to the findings?

3. What was the proportion of missing data overall? Could that information be added in the discussion?

4. The checklist in the supplementary data is missing page numbers after third page and needs to be completed.

6. PLOS authors have the option to publish the peer review history of their article (what does this mean?). If published, this will include your full peer review and any attached files.

Reviewer #1: No

Reviewer #2: No

---

## [Author Response · Author response to Decision Letter 0]

3 Sep 2021

See rebuttal letter for response to reviewers comments

---

## [Editor Report · Decision Letter 1]

27 Oct 2021

Changes in PHQ-9 Depression Scores in Acute Stroke Patients Shortly After Returning Home

PONE-D-21-15011R1

Dear Dr. Reeves,

We’re pleased to inform you that your manuscript has been judged scientifically suitable for publication and will be formally accepted for publication once it meets all outstanding technical requirements.

Kind regards,

Seana Gall

Academic Editor

PLOS ONE

---

## [Editor Report · Acceptance letter]

2 Nov 2021

PONE-D-21-15011R1 

Changes in PHQ-9 depression scores in acute stroke patients shortly after returning home 

Dear Dr. Reeves:

I'm pleased to inform you that your manuscript has been deemed suitable for publication in PLOS ONE. Congratulations! Your manuscript is now with our production department. 

Kind regards, 

on behalf of

Dr. Seana Gall 

Academic Editor

PLOS ONE